# Effects of Roxadustat on Erythropoietin Production in the Rat Body

**DOI:** 10.3390/molecules27031119

**Published:** 2022-02-08

**Authors:** Yukiko Yasuoka, Yuichiro Izumi, Takashi Fukuyama, Haruki Omiya, Truyen D. Pham, Hideki Inoue, Tomomi Oshima, Taiga Yamazaki, Takayuki Uematsu, Noritada Kobayashi, Yoshitaka Shimada, Yasushi Nagaba, Tetsuro Yamashita, Masashi Mukoyama, Yuichi Sato, Susan M. Wall, Jeff M. Sands, Noriko Takahashi, Katsumasa Kawahara, Hiroshi Nonoguchi

**Affiliations:** 1Department of Physiology, Kitasato University School of Medicine, 1-15-1 Kitasato, Minami-ku, Sagamihara 252-0374, Kanagawa, Japan; yasuoka@med.kitasato-u.ac.jp (Y.Y.); tomomio@kitasato-u.ac.jp (T.O.); ntakahas@med.kitasato-u.ac.jp (N.T.); kawahara@kitasato-u.ac.jp (K.K.); 2Department of Nephrology, Kumamoto University Graduate School of Medical Sciences, 1-1-1 Honjo, Chuo-ku, Kumamoto 860-8556, Kumamoto, Japan; izumi_yu@kumamoto-u.ac.jp (Y.I.); hideki24@kumamoto-u.ac.jp (H.I.); mmuko@kumamoto-u.ac.jp (M.M.); 3Division of Biomedical Research, Kitasato University Medical Center, 6-100 Arai, Kitamoto 364-8501, Saitama, Japan; fukuyam@insti.kitasato-u.ac.jp (T.F.); tyamazak@insti.kitasato-u.ac.jp (T.Y.); tuematsu@insti.kitasato-u.ac.jp (T.U.); kenchu@insti.kitasato-u.ac.jp (N.K.); 4Department of Biological Chemistry and Food Sciences, Faculty of Agriculture, Iwate University, 3-18-8 Ueda, Morioka 020-8550, Iwate, Japan; ommandola@gmail.com (H.O.); yamashit@iwate-u.ac.jp (T.Y.); 5Renal Division, Department of Medicine, Emory University School of Medicine, 1639 Pierce Drive, WMB Room 3313, Atlanta, GA 30322, USA; derek.pham@emory.edu (T.D.P.); smwall@emory.edu (S.M.W.); jeff.sands@emory.edu (J.M.S.); 6Division of Internal Medicine, Kitasato University Medical Center, 6-100 Arai, Kitamoto 364-8501, Saitama, Japan; yoshi@insti.kitasato-u.ac.jp (Y.S.); nagaba-y@insti.kitasato-u.ac.jp (Y.N.); 7Department of Molecular Diagnostics, Kitasato University School of Allied Health Sciences, Sagamihara 252-0373, Kanagawa, Japan; yuichi@med.kitasato-u.ac.jp

**Keywords:** erythropoietin, PHD inhibitor, Roxadustat, hypoxia, deglycosylation, Western blotting, HIF2α, proximal tubules, collecting ducts

## Abstract

Anemia is a major complication of chronic renal failure. To treat this anemia, prolylhydroxylase domain enzyme (PHD) inhibitors as well as erythropoiesis-stimulating agents (ESAs) have been used. Although PHD inhibitors rapidly stimulate erythropoietin (Epo) production, the precise sites of Epo production following the administration of these drugs have not been identified. We developed a novel method for the detection of the Epo protein that employs deglycosylation-coupled Western blotting. With protein deglycosylation, tissue Epo contents can be quantified over an extremely wide range. Using this method, we examined the effects of the PHD inhibitor, Roxadustat (ROX), and severe hypoxia on Epo production in various tissues in rats. We observed that ROX increased Epo mRNA expression in both the kidneys and liver. However, Epo protein was detected in the kidneys but not in the liver. Epo protein was also detected in the salivary glands, spleen, epididymis and ovaries. However, both PHD inhibitors (ROX) and severe hypoxia increased the Epo protein abundance only in the kidneys. These data show that, while Epo is produced in many tissues, PHD inhibitors as well as severe hypoxia regulate Epo production only in the kidneys.

## 1. Introduction

Anemia is common world-wide [1] and is a major complication of chronic renal failure [2]. Hypoxia and anemia both stimulate Epo production in the kidneys [3,4,5] by inhibiting the Prolyl Hydroxylase Domain (PHD), which then stimulates hypoxia-inducible factor (HIF)1α/2α [6]. PHD inhibitors mimic severe hypoxia, thereby, reducing HIF1α/2α degradation, which increases HIF1α/2α activity and Epo production [7,8,9,10,11,12,13,14]. Treatment of the anemia of chronic kidney failure was revolutionized with the development of human recombinant erythropoietin (Epo) and erythropoiesis stimulating agents (ESAs) [4].

Whereas HIF1α is produced in many tissues, HIF2α is expressed only in lung and epithelial cells [15,16,17,18]. PHD inhibitors stimulate HIF production in these other tissues, which has benefits beyond the treatment of anemia, including a reduction in myocardial infarction, atherosclerosis and peripheral artery disease [19]. However, this HIF1α/2α stimulation also promotes cancer progression as well as other cardiac and metabolic diseases [20,21,22,23,24,25].

We therefore hypothesized that, if these drugs increase Epo by targeting the prolylhydroxylase domain enzyme outside the kidneys, then this extrarenal increase in HIF1α/2α expression might induce unwanted side effects. Thus, a long-term increase in HIF1α/2α expression outside the kidneys may invoke other diseases. As such, the purpose of this study was to determine the tissue(s) in which Epo production increases with prolylhydroxylase domain enzyme inhibitor (ROX) administration. To accomplish this goal, we developed a novel assay that employs immunoblots of deglycosylated tissue samples, and this enables Epo protein abundance to be quantified over a wide range.

The hematopoietic effects of PHD inhibitors on renal anemia in chronic kidney disease (CKD) patents have been established [9,26,27]. Whether PHD inhibitors stimulate Epo production in the liver or not is the major question. Severe hypoxia and PHD inhibition are known to stimulate Epo mRNA expression not only in the kidneys but also in the liver [28,29,30]. Kato et al. reported that TP0463518, a PHD inhibitor, stimulated Epo mRNA expression in the liver but not in the kidneys in 5/6 nephrectomized rats [29], suggesting that TP0463518 increases Epo production in the liver. 

Moderate hepatic impairment did not affect the hematopoietic effects of Roxadustat (ROX), a PHD inhibitor, suggesting a small role of the liver for Epo production by ROX [31]. A major problem in these studies was the lack of Epo protein determination. They examined the Epo mRNA and plasma Epo concentrations to evaluate the Epo protein production. The liver produces Epo in utero but then stops at birth through Epo gene methylation [32,33,34]. These data suggest that, within the liver, Epo transcription occurs but does not result in protein translation. 

To test this hypothesis, a better approach to quantifying Epo abundance in tissue and in plasma is required. To accomplish this goal, we developed a method for quantifying Epo tissue that employs Western blot analysis and does not require pre-purification of the samples [35,36]. We observed that Epo deglycosylation with PNGase increases Epo assay sensitivity ~10 fold. Using this Epo assay, we investigated which tissues increase Epo production in response to the PHD inhibitor, ROX or in response to severe hypoxia.

## 2. Results

### 2.1. Comparison of Glycosylated and Deglycosylated Epo Protein by Western Blotting

We compared the glycosylated and deglycosylated Epo protein abundance using Western blotting. ROX-treated kidney lysates were deglycosylated with or without PNGase and were examined by two-fold serial dilution. As shown, the threshold of detection was lower for the deglycosylated compared with for the glycosylated protein (Figure 1).

### 2.2. Western Blot Analysis of ROX-Induced Epo Production in the Body

ROX-induced Epo production in the body was examined using deglycosylated Epo. Epo production in ROX-treated male rats was observed most abundantly in the kidneys (Figure 2a). Low levels of Epo protein abundance were seen in the salivary glands, pancreas and epididymis. Therefore, we tested whether ROX stimulates Epo production in these tissues by comparing the Epo abundance by immunoblot of lysates from control and ROX-treated rats. 

We observed, however, that Epo protein abundance in the salivary glands, pancreas, and epididymis were unchanged with ROX treatment (Figure 2b,c). Epo protein expression was not detected in the liver from either control or ROX-treated rats (Figure 2a). ROX treatment increased the Epo protein abundance only in the kidneys (Figure 2b). The effect of ROX on Epo protein expression was also examined in female rats. Small expressions were observed in the salivary glands, thymus, spleen and ovaries. ROX did not increase the expression in these organs (Figure 2d,e). The increase of Epo protein expression was observed only in the kidneys, which was the same for males. Our findings are summarized in Figure 2f.

### 2.3. Epo Production during Severe Hypoxia (In Vivo)

The next series of experiments examined the effect of severe hypoxia on Epo production in various tissues taken from male and female rats. In response to hypoxia, the Epo protein abundance increased dramatically in the kidneys from both male and female rats (Figure 3b,c,e). A low level of Epo protein was detected in the salivary glands, pancreas, spleen and epididymis or ovaries in males and females, although the Epo protein abundance was unchanged in these tissues in response to severe hypoxia (Figure 2a–e). Our findings are summarized in Figure 3f.

### 2.4. Effects of ROX on Epo mRNA and Protein Expression in the Kidneys and Liver

Epo mRNA expression was investigated in the renal cortex and liver. As shown, ROX increased Epo mRNA in both the renal cortex and liver (kidney: 1, 1143 ± 445* and 718 ± 331* and liver: 1, 236 ± 48* and 327 ± 164* fold in control, R5 and R10, respectively, Epo mRNA/β-actin mRNA expression of control kidney or liver was considered as 1 (n = 3–7, Figure 4a). Moreover, in response to ROX treatment, HIF2α mRNA increased in both the renal cortex and the liver (kidney: 1, 7.3 ± 3.3* and 11.2 ± 6.5* and liver: 1, 1.7 ± 0.3 and 1.8 ± 0.1* fold in control, R5 and R10, respectively, n = 3–6, Figure 4b). 

However, while ROX increased HIF1α mRNA in the kidneys, ROX reduced HIF1α mRNA in the liver (kidney: 1, 9.0 ± 5.6* and 14.8 ± 8.5* and liver: 1, 0.72 ± 0.06* and 0.76± 0.01* fold in control, R5 and R10, respectively, n = 3–5, Figure 4c). By the administration of the PHD inhibitor, ROX, PHD2 mRNA expression decreased in the renal cortex but increased in the liver (kidney: 1, 0.78 ± 0.22 and 0.25 ± 0.02* and liver:1, 5.38 ± 0.92* and 11.83 ± 1.31* fold in control, R5 and R10, respectively, n = 4–8, Figure 4d). We showed previously that, during severe hypoxia, Epo mRNA in the liver was only 1.5% of that seen in the kidneys. 

Nonetheless, Epo mRNA expression in the liver increases during severe hypoxia (1, 36.7 ± 10.0*, 62.8 ± 12.0* and 183.0 ± 16.2* fold in 0, 1, 2 and 4 h after hypoxia, respectively, n = 3–5, * *p* < 0.05).

In response to ROX, we observed an increase in the 35–38 kDa band density in lysates from the kidneys (lane R10) and liver (lanes 3 and 10, respectively). However, in response to ROX, while the 22 kDa band density increased in deglycosylated samples from the renal cortex (Figure 4e), no 22 kDa band was observed in deglycosylated samples from the liver (Figure 4e). The absence of a 22 kDa band in the deglycosylated liver lysates indicates that Epo was not produced by this tissue. These results also show that Epo production can be accurately quantified by measuring the abundance of the 22 kDa band by immunoblot of deglycosylated tissue samples.

### 2.5. Immunohistochemistry (IHC) of Epo Production by the Kidney

While Epo staining was seen in proximal and distal tubules in control rats, the label intensity did not change significantly with ROX administration (Figure 5g–i). Only a small increase in label intensity was observed in the proximal tubule following ROX administration (Figure 5g–i). 

Instead, ROX administration produced the greatest change in the Epo label within interstitial cells. While Epo staining was not detected in peritubular cells under basal conditions (Figure 5a,d), ROX increased the Epo label of the peritubular cells that surround the proximal tubule in a dose-dependent manner (Figure 5b,c,e,f).

### 2.6. Plasma Epo Concentration

Plasma Epo concentrations in control and ROX-treated rats are shown in Figure 6a. ROX (50 mg/kg) significantly increased the plasma Epo concentration from 1.2 ± 0.1 to 1072 ± 333 mIU/mL (n = 5–6, *p* < 0.001). The plasma Epo expression was also estimated using Western blotting. ROX increased the plasma Epo concentrations (1.4, 2690 and 3320 mIU/mL in control, R5 and R10, respectively). By Western blot of the plasma Epo, ROX dose-dependently increased both the glycosylated and deglycosylated Epo protein (Figure 6b). The abundance of the deglycosylated Epo protein in sample R10 was slightly less than that that of 498 pg of recombinant rat Epo protein.

## 3. Discussion

The purpose of this study was to define the sites of Epo production both under basal conditions and in response to ROX or hypoxia. Therefore, we investigated Epo protein abundance under basal conditions and following either severe hypoxia or ROX treatment in many tissues, including the cerebrum, cerebellum, salivary glands, thymus, lungs, heart, liver, spleen, pancreas, adrenal glands, kidneys, testis, epididymis and ovaries. 

While we detected Epo protein in the salivary glands, spleen, epididymis and ovaries under basal conditions, neither ROX nor severe hypoxia stimulated Epo production in those organs. Instead, our data showed that ROX and hypoxia stimulated Epo production only in the kidneys. Although severe hypoxia and ROX stimulated Epo mRNA expression in the liver, Epo protein production was not detected in the liver. Although ROX and hypoxia-induced Epo production was not observed other than in the kidneys, we cannot exclude the possibility that ROX produces side effects by targeting PHD in other tissues.

Epo and HIF2α mRNA expression increased not only in the kidneys but also in the liver. HIF1α mRNA expression was increased in the kidneys but decreased in the liver. PHD2 mRNA expression was decreased in the kidneys but increased in the liver. These data strongly suggest that the PHD inhibitor, ROX, stimulates Epo production through its effect in the kidneys but not in the liver.

To properly measure the Epo protein expression, we used both glycosylated and deglycosylated Epo expression. We showed the lower detection limit of deglycosylated 22 kDa bands compared with glycosylated 34–38 kDa bands using ROX-treated kidney lysates in this study (Figure 1c). Deglycosylated Epo production was not observed in the liver even with the increase of Epo mRNA expression. DNA methylation after birth may cause the lack of Epo production after birth and under severe hypoxia or under PHD inhibition by ROX [32,33,34]. These data suggest the importance of measuring the Epo protein but not Epo mRNA expression.

In previous studies, we showed that Epo production is regulated not only by hypoxia but also by the renin-angiotensin-aldosterone system [35,37,38]. We previously showed that fludrocortisone and angiotensin II stimulated Epo production by the nephron, especially in the intercalated cells of the collecting duct [35,37,38]. 

In contrast, severe hypoxia and ROX stimulate Epo production through renal erythropoietin producing (REP) cells [34,39,40,41,42,43,44] and increase the plasma Epo concentration. Our immunohistochemistry showed that both ROX and severe hypoxia increase Epo production by interstitial cells but not by tubular cells. The increase of plasma Epo concentration by ROX or severe hypoxia was higher than that by fludrocortisone or angiotensin II, showing that the Epo producing ability of REP cells is much higher than that of nephrons.

## 4. Materials and Methods

### 4.1. Materials and Animals

Male Sprague Dawley rats (Japan SLC, Hamamatsu, Japan) were used in our study. The rats were divided into four groups: 1, control (six rats); 2, ROX-treatment (nine rats); 3, control to hypoxia (six rats); and 4, severe hypoxia (six rats). The ROX-group rats were given an intraperitoneal injection of ROX (FG-4592; MedChemExpress, Monmouth Junction, NJ, USA at 50 or 100 mg/kg body weight (R5 and R10, respectively), dissolved in 5% glucose with 32.5 mmol/L NaOH) [45]. The control group was injected with vehicle (5% glucose with 32.5 mmol/L NaOH). 

After 6 h, rats were injected with mixed anesthetic (0.3 mg/kg of medetomidine, 4.0 mg/kg of midazolam and 5.0 mg/kg of butorphanol), and blood was taken from the heart. The hypoxia-group rats were placed in 7% O_2_ for 4 h, and control rats were placed in room air for 4 h before the injection of mixed anesthetic. Organs (the olfactory bulb, cerebral cortex, cerebellum, salivary glands, thymus, lungs, heart, liver, spleen, pancreas, adrenal glands, kidney cortex, epididymis and ovaries were taken after perfusing 20 mL of PBS from the heart. Our protocols were checked and approved by the Ethics Committee at Kitasato University Medical Center (25-2, 2018032, 2019029) and Kitasato University School of Medicine (2020-042).

### 4.2. Western Blot Analysis with Enzymatic Deglycosylation

Deglycosylation-coupled Western blot analysis was performed as described previously [26,31,32]. For the Western blot of plasma Epo, 10 μL of plasma was subjected to deglycosylation with and without PNGase as described below. Half of each sample was used for Western blot analysis. For the Western blot of tissue Epo, protein was extracted from organs using CelLytic MT (C-3228; Sigma-Aldrich, Burlington, VT, USA) plus protease inhibitor (05892970001, Roch, Basel, Switzerland) and used for Western blotting. In certain experiments, plasma from control and ROX-treated rats was also used. 

Samples (lysates or plasma) were deglycosylated using N-glycosidase F (PNGase, 4450; Takara Bio, Kusatsu, Japan). We added 1 μL of 10% SDS to 10 μL of lysates samples and boiled for 3 min. Then, 11 μL of 2× stabilizing buffer was added, and the samples were vortexed. After the addition of 1 μL of PBS (glycosylated Epo) or PNGase (deglycosylated Epo), the samples were incubated in a water bath for 17–20 h at 37 °C. After the incubation, the samples were spun down, and the supernatant was collected and used for SDS-PAGE (10–20% gradient gel, 414893; Cosmo Bio, Tokyo, Japan). 

The 2× stabilizing buffer contained 125 mM Tris-HCl (pH 8.6), 48 mM EDTA, 4% Nonidet P-40 and 8% 2-mercaptoethanol. Recombinant rat Epo (rRatEpo, 592302; BioLegend, San Diego, CA, USA) was used as a positive control in both glycosylated and deglycosylated samples. After SDS-PAGE, proteins were transferred to a 0.45 μm PVDF membrane (Immobilon-P, IPVH00010; Merck Millipore, Burlington, VA, USA) with 120 mA for 60–90 min. 

The membrane was blocked with 5% skim milk (Morinaga, Tokyo, Japan) for 60 min and incubated with the antibody against Epo (sc-5290, 1:500; Santa Cruz) for 60 min at room temperature. After washing, the membrane was incubated with a secondary antibody (goat anti-mouse IgG (H+L) (115-035-166, 1:5000; Jackson ImmunoResearch Laboratorie, West Grove, USA) for 60 min. 

Bands were visualized by the ECL Select Western Blotting Detection System (RPN2235; GE Healthcare Bio-Science AB, Uppsala, Sweden) and LAS 4000 (Fujifilm, Tokyo, Japan). Densitometric analysis was performed by Multi Gauge in LAS 4000. After measuring the Epo protein expression, the membrane was stripped (stripping solution, Wako, RR39LR, Tokyo, Japan) and reprobed with the antibody against β-actin (MBL, M177-3, Tokyo, Japan) for the normalization of the band. The molecular weight marker used was PagaRuler (26616, Thermo Scientific, Waltham, MA, USA).

The band at 35–38 kDa does not guarantee that the band is Epo. The band shift from 35–38 to 22 kDa by PNGase guarantees that the band is deglycosylated Epo protein. We showed that the 35–38 kDa and shifted to 22 kDa bands detected by sc-5290 represent Epo by LC/MS analysis using cut gels [35,36]. The assay sensitivity of glycosylated and deglycosylated Epo was compared with ROX-treated kidney lysates.

### 4.3. Real Time Quantitative RT-PCR

RNA was extracted from the kidneys and liver using Qiacube and the RNeasy Mini Kit (74106; Quiagen, Venlo, Netherland) as described previously ([35,38]. cDNA was synthesized using a Takara PrimeScript II 1st strand cDNA Synthesis Kit (6210; Takara Bio). Real Time PCR was performed using probes from Applied Biosystems, Waltham, MA, USA (β-actin Rn00667869_m1, Epo Rn00667869_m1, HIF2α Rn00576515_m1, HIF1α Rn01472831_m1 and PHD2 Rn00710295_m1) and Premix Ex Taq (RP39LR; Takara Bio). The mRNA expression in control and ROX rats were compared by relative gene expression data using real-time quantitative PCR and the 2-ΔΔCT by Livak K.J., et al. [46].

### 4.4. Immunohistochemistry

Kidney sections were immuno-stained as described previously [35,37,38,47]. In brief, the sections were blocked with 5% normal goat serum and reacted with rabbit polyclonal anti-human Epo antibody (sc-7956, 1:10; Santa Cruz Biotechnology, Santa Cruz, CA, USA), followed by Histofine Simple Stain MAX-PO (414341F; Nichirei Bioscience, Tokyo, Japan). Sections were stained using DAB liquid system (BSB 0016; Bio SB, Santa Barbara, CA, USA) and counterstained with Mayer’s hematoxylin (30002; Muto Pure Chemicals, Tokyo, Japan).

Images were obtained using an optical microscope (Axio Imager M2; Carl Zeiss, Oberkochen, Germany) with a digital camera (AxioCam 506, Carl Zeiss). Captured images were analyzed using an image analyzing system (ZEN 2, Carl Zeiss).

### 4.5. Plasma Epo Concentration Measurements

Plasma samples were taken from control and ROX-treated rats at 6 h after peritoneal injection. Plasma Epo concentrations were measured by CLEIA (SRL, Tokyo, Japan).

### 4.6. Statistical Analyses

Data are expressed as the mean ± SEM. Statistical significance was performed using Excel Statics (BellCurve, Tokyo, Japan). Statistical significance was analyzed using non-parametric analysis of the Kolmogorov–Smirnov test, Wilcoxon signed rank test or the Kruskal–Wallis test and multiple comparisons by the Shirley–Williams test. *p* < 0.05 was considered statistically significant.

## 5. Conclusions

In conclusion, our study showed that ROX and severe hypoxia increased Epo production only by the kidney interstitial cells by stimulating HIF1α and 2α expression and inhibiting PHD2 expression. Although ROX and severe hypoxia increased the Epo mRNA expression in the liver, Epo protein production was not observed.

## Figures and Tables

**Figure 1 molecules-27-01119-f001:**
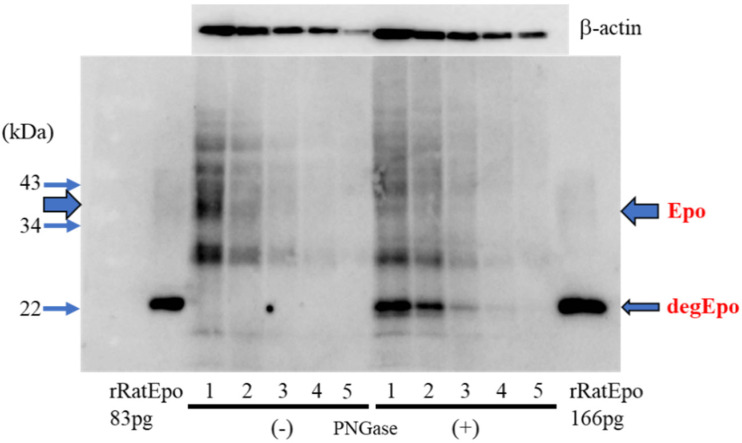
Comparison of glycosylated and deglycosylated Epo proteins by Western blotting. Lanes 1–5 show the same amount of glycosylated (PNGase (−)) and deglycosylated (PNGase (+)) Epo. Epo; (glycosylated) erythropoietin, degEpo; deglycosylated Epo, rRatEpo; recombinant rat Epo. Glycosylated and deglycosylated recombinant rat Epo were combined.

**Figure 2 molecules-27-01119-f002:**
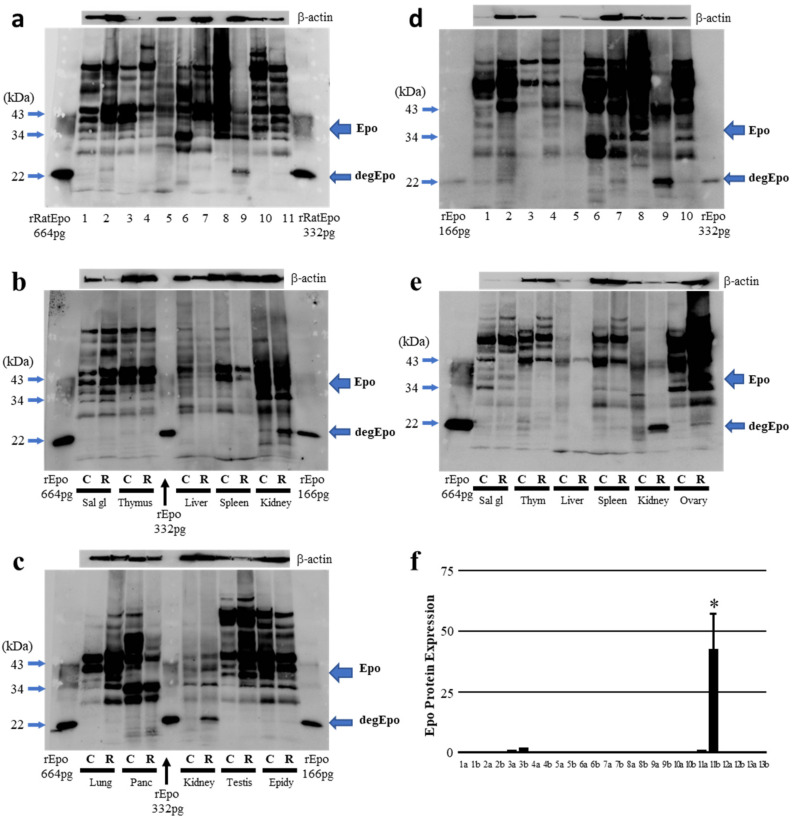
Western blot analysis of ROX-induced Epo production in the body. (**a**) Epo protein expression in ROX-treated male rat. 1, Salivary gland; 2, thymus; 3, lung; 4, heart; 5, liver; 6, pancreas; 7, spleen; 8, adrenal gland; 9, kidney (cortex); 10, testes; and 11, epididymis. (**b**) Epo protein expressions in salivary gland, thymus, liver, spleen and kidney were compared in control and ROX-treated male rats. (**c**) Epo protein abundances in lung, pancreas, kidney, testis and epididymis were compared in control and ROX-treatedmale rats. (**d**) Epo protein expression in ROX-treated female rat. 1, Salivary gland; 2, thymus; 3, lung; 4, heart; 5, liver; 6, pancreas; 7, spleen; 8, adrenal gland; 9, kidney (cortex); and 10, ovary. (**e**) Epo protein expressions in salivary gland, thymus, liver, spleen, kidney and ovaries were compared in control and ROX-treated female rats. (**f**) Summarized data. Epo protein expression was normalized by measuring the β-actin expression. * indicates *p* < 0.05, n = 3–6. 1, Olfactory bulb; 2, cerebrum; 3, salivary gland; 4, thymus; 5, lung; 6, heart; 7, liver; 8, pancreas; 9, spleen; 10, adrenal gland; 11, kidney (cortex); 12, epididymis; and 13, ovary. a, Control and b, ROX.

**Figure 3 molecules-27-01119-f003:**
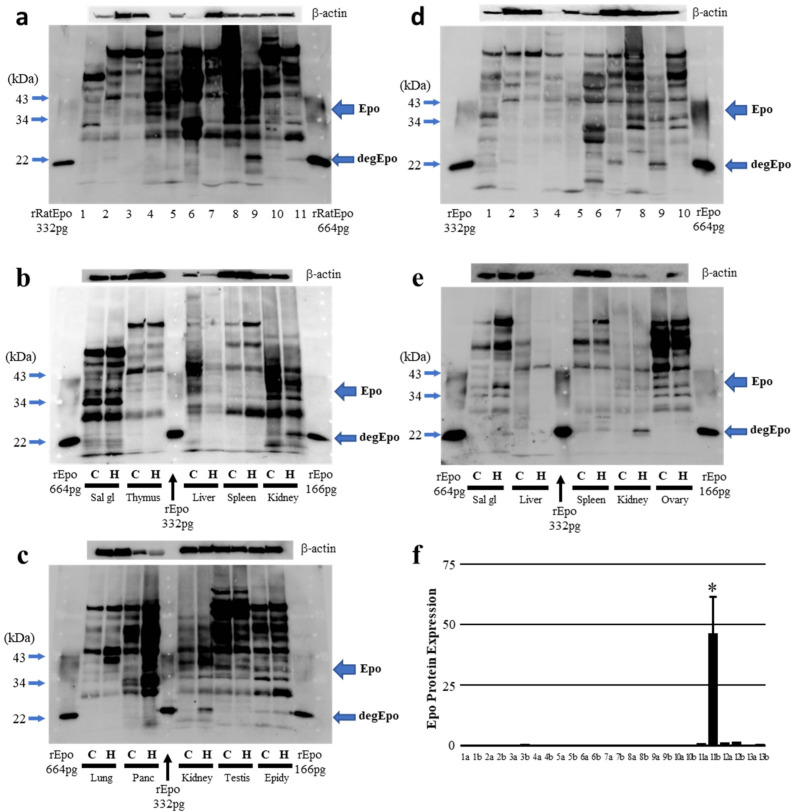
Effects of severe hypoxia on Epo protein expression in the body. (**a**) Epo protein expression in severe hypoxic male rat. (**b**) Epo protein expressions in salivary gland, thymus, liver, spleen and kidney were compared in control and severe hypoxic male rats. (**c**) Epo protein abundances in lung, pancreas, kidney, testis and epididymis were compared in control and severe hypoxic rats. (**d**) Epo protein expression in severe hypoxic female rat. (**e**) Epo protein expressions in salivary gland, liver, spleen, kidney and ovaries were compared in control and severe hypoxic female rats. (**a**,**d**) 1–10, the same as in Figure 2. (**f**) Summarized data. Statistical significance between control and hypoxic-treatment was obtained only in the kidneys. * indicates *p* < 0.05, n = 3–6. 1–13, the same as in Figure 2. a, control and b, severe hypoxia.

**Figure 4 molecules-27-01119-f004:**
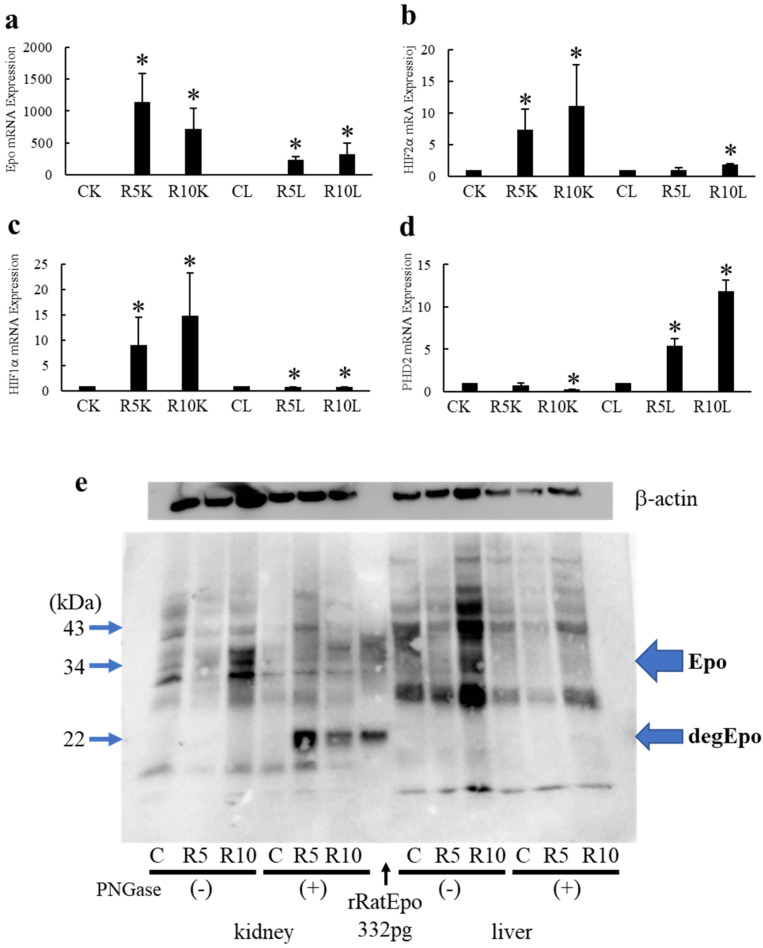
Effects of ROX on Epo mRNA and protein expression in the kidneys and liver. Effects of ROX on Epo (**a**), HIF2α (**b**), HIF1α (**c**) and PHD2 (**d**) mRNA expressions. (**e**) Effects of ROX on Epo protein expressions in the kidney (left half) and the liver (right half). * indicates *p* < 0.05. n = 3–9. CK, R5K and R10K show the control, R5- and R10-treated rat kidney, respectively. CL, R5L and R10L show the control, R5- and R10-treated rat liver, respectively.

**Figure 5 molecules-27-01119-f005:**
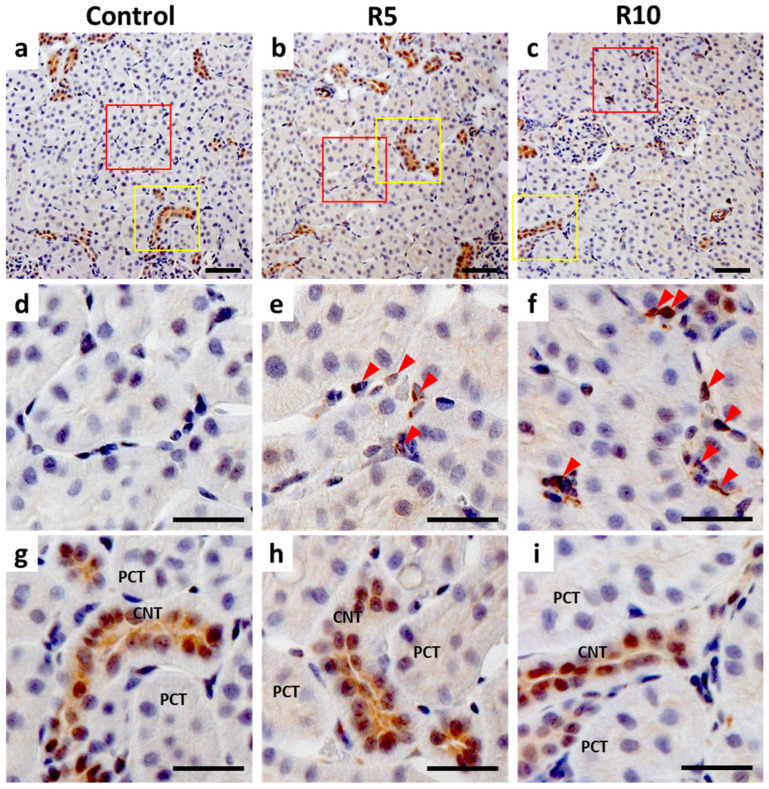
Immunohistochemical analysis of the effects of ROX on Epo production by the kidneys. Under control conditions, Epo staining was found from the proximal tubule to the collecting duct (**a**). ROX increased Epo production in the peritubular cells around proximal tubules after 6 h (R10 > R5). Red arrowheads show Epo positive peritubular cells ((**d**–**f**) and red square of (**a**–**c**)). Epo staining was not increased in the distal tubules but slightly increased in the proximal tubules by ROX ((**g**–**i**) and yellow square of (**a**–**c**)). Scale bar: 20 µm (**a**–**c**) and 10 µm (**d**–**i**). PCT, proximal convoluted tubules; and CNT, connecting tubules.

**Figure 6 molecules-27-01119-f006:**
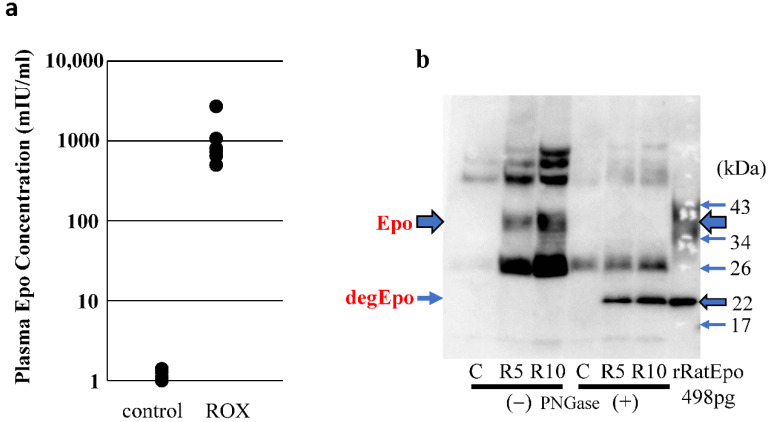
Plasma Epo concentration and Western blot analysis of serum Epo. (**a**): Plasma Epo concentration in control and ROX (50 mg/kg)-injected rats. ROX significantly increased the plasma Epo concentration from 1.2 ± 0.1 to 1072 ± 333 mIU/mL (n = 5–6, *p* < 0.001). (**b**): Serum Epo was examined by Western blot analysis as described in the methods. Plasma Epo concentration in control, R5 (50 mg/kg) and R10 (100 mg/kg) rats was 1.4, 2696 and 3200 mIU/mL, respectively. Glycosylated Epo (Epo) and deglycosylated Epo (degEpo) were detected at 35–38 and 22 kDa, respectively.

## Data Availability

The data presented in this study are openly available.

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
