# Peer review of "Effects of Roxadustat on Erythropoietin Production in the Rat Body"

_molecules, 2022, doi:10.3390/molecules27031119_

Round 1
Reviewer 1 Report
Dear Authors,
The manuscript is interesting and has great practical potential. But the presentation of results needs to be reworked. I have major concerns outlined below.
To Abstract. Not specified about the object of research. It should be noted that the results were obtained in rats.
To Introduction. It also does not indicate which objects were used for experiments. It is necessary to describe what ROX is and what data have already been obtained by other authors on the effectiveness of its use for the production of Epo and at what objects, If there are any. Besides, please indicate what are the advantages of your proposed method for detecting Epo in comparison with others, If there are any. Has Epo concentration been previously detected in human blood samples in the same manner?
This section does not need to describe the results. Here you only need to indicate which hypothesis you specifically proposed to test. And further “To accomplished this goal, we developed a method for quantifying tissue Epo that 62 employs Western blot analysis that does not require pre-purification of samples [26, 31]. We observed that 63 Epo deglycoslyation with PNGase increases Epo assay sensitivity ~ 10-fold. Using this Epo assay, we 64 investigated which tissues increase Epo production in response to the PHD inhibitor, Roxadustat (ROX) or 65 in response to severe hypoxia "
To Results. In general, this section needs to be rewritten to clearly present only the results and not discuss them. A discussion of the results should be presented in the “Discussion”. In addition, the order of presentation of the results should be revised. From my point of view, it is more logical to present an increase in the concentration of the Epo protein in organs (males and females) at first, then the detection of the Epo protein and mRNA in the kidneys and liver, then histochemistry, and the concentration of Epo in the plasma at the end.
- «Plasma Epo concentration»
This section does not need to provide details (SRL, Tokyo, Japan) and (by Kolmogorov – Smirnov test). You have already indicated all these details in the description of the Methods section.
The legend to Figure 1 should be concise and not repeat the description of the results presented above.
It is not entirely clear why plasma is used to determine Epo concentration using the CLEIA assay, while serum is used for Western blotting.
It is not entirely clear why in the “Plasma Epo concentration” section presents a blot with an ROX-treated kidney. It is more logical to indicate the results for tissues in the appropriate paragraphs, which will be indicated at the beginning, as I suggested above
- «Effects of ROX on Epo-related mRNA expression in kidney and liver»
Doesn’t need to provide details in this section (by the Kruskal-110 Wallis test and multiple comparisons by the Shirley-Williams test).
The legend to Figure 2 should be concise and not repeat the description of the results presented above.
It is not entirely clear why Fig. 2f is shown, while its description is not in the results. Moreover, the legend to the figure should not include a discussion of the results.
Instead of specifying the * character in the text next to the numbers, you must specify the p-value.
This proposal “However, we have shown previously 111 that during severe hypoxia, Epo mRNA in the liver is only 1.5% of that seen in the kidney. We conclude that 112 while hypoxia stimulates Epo mRNA in both liver and kidney, Epo mRNA expression remains much lower 113 in the former relative to the latter tissue. " should be indicated in the discussion.
Please provide a link to the source in the sentence "However, we have shown previously 111 that during severe hypoxia, Epo mRNA in the liver is only 1.5% of that seen in the kidney."
- Western blot analysis of ROX-induced Epo production in kidney and liver
Why does this section have refer to Figure 2e, which is shown in the section above and not relevant to this section?
I don't quite understand what is meant by "band density"? Have you normalized to β-actin? If yes, then it is necessary to include these histograms with an indication of the significance value
- Western blot analysis of ROX-induced Epo production in the body
In this section, it is also necessary to present only the figures, the description of which follows from the section, in order to avoid confusion.
In this phrase «In the liver, Epo protein expression was not detected either under basal conditions or with ROX 153 treatment. (Fig. 4a). Instead, ROX increased Epo protein abundance only in the kidney (Fig. 4b).» you give a description of the results on the concentration of Epo in females, however, this mention is not made in the text.
In the figures for this section, it is necessary to indicate only the decoding of abbreviations. Results should not be described in figure legends.
In addition, I propose to combine Figures 3 and 4, placing close Fig. 3a and 4a, 3b and 4b, and etc. to see the big picture for males and females.
- Epo production during severe hypoxia (in vivo)
In this section, it is also necessary to combine the figures by analogy with the above indicated. In addition, it is necessary to combine the description of the results for males and females, since the results are similar and should be presented briefly for clear understanding.
Concerning to fig. 7. What was the normalization done? The legend under the figure is not entirely clear. If this histogram is to demonstrate statistical significance for the concentration of Epo in different organs, then it should be placed next to the blots, and not at the end.
- Immunohistochemistry (IHC) of Epo production by the kidney
The phrase “Our findings are summarized in Fig. 7a and 7b. In the rat, severe hypoxia and ROX both stimulate Epo 208 production” in this section cannot be given, since the figure is described above. And the description of results referencing the figure should always follow before the figure.
To Discussion
The phrase «PHD inhibitors are used to treat patients with renal anemia [7, 9-14, 32]. One of the benefits of the PHD 223 inhibitors is that they can be administered by mouth, whereas ESAs require subcutaneous injection. While 224 PHD inhibitors are clearly of benefit in the treatment of anemia, they are limited by the malignancy and 225 inflammation [20-25] that accompanies the widespread HIF1α/2α stimulation induced by these drugs.» already presented in the Introduction. The description should not be repeated.
In this phrase "Deglycosylated Epo production was not observed in the liver even 241 with the increase of Epo mRNA expression." should include a description of what is Deglycosylated Epo.
In this phrase «In previous studies, we showed that Epo production is regulated not only by hypoxia but also by the 244 renin-angiotensin-aldosterone system [26, 32, 33]. Severe hypoxia stimulates Epo production by renal 245 interstitial cells, which we call renal Epo producing (REP) cells [30, 34-39]. In contrast, fludrocortisone, an 246 analogue of aldosterone, and angiotensin II (ATII), stimulate Epo production by the cortical nephron, 247 especially by intercalated cells [32, 33].» the description should be shortened if these data have already been described earlier.
This description should be given in the Materials section “ Western blot analysis with enzymatic deglycosylation.“ And in the Discussion, already give only those data that are obtained in this study, indicating a link to the previous study - «As we described in the results section, the band at 35-38 kDa does not guarantee 250 that the band is Epo. The band shift from 35-38 to 22 kDa by PNGase guarantees that the band is 251 deglycosylated Epo protein. We showed that the 35-38 kDa and shifted to 22 kDa bands detected by sc-5290 252 represent Epo by LC/MS analysis using cut gels [26, 31].»
To Materials
At the beginning of this section, I propose to present a “flowchart of the study”, indicating the methods, groups with the number and type of samples that were used to better understanding the stages of the experiment.
- Why was the CLEIA method used instead of ELISA to determine the Epo concentration? What are its advantages over ELISA and are there any publications on the use of this method? It is possible then to provide a link to this. What test system was used for determing of Epo?
- Since you determine the expression of mRNA, then it is more correct to use the name "Real-Time Quantitative RT-PCR"
- What threshold level of expression (Ct) was used? Please indicate.
- mRNA expressions in control and ATII rats were compared with ΔΔCT. What is meant? Do you mean the method by Livak et al.? (Analysis of Relative Gene Expression Data Using Real-Time Quantitative PCR and the 2 − ΔΔCT Method). If so, then it would be more correct to write as “The level of mRNA expression in control and ATII rats was determined by the 2 − ΔΔCT method (indicate the author link) I don't quite understand in which experiment you used ATII rats?
- In the section “Western blot analysis with enzymatic deglycosylation” should clearly indicate which samples were used for the Western blot with enzymatic deglycosylation and which were used for the classical Western blot.
- What is the thickness of the PVDF membrane you used?
- What molecular weight marker did you use?
- What reference protein was used for tissue normalization?
- What was normalized for in plasma?
And please indicate which software was used for the Densitometric analysis
In the phrase “After SDS-PAGE, proteins were transferred to a PVDF membrane (Immobilon-P, 302 IPVH00010; Merck Millipore, Burlington, USA) with 120 mA for 60-90 min” it is not clear under what conditions the transfer from the gel to the membrane. If I understand correctly, then 120 mA for 60-90 min are the conditions for protein denaturation, but not for transfer to the membrane. Is it right?

Author Response
Answers to the Reviewer 1
We thank Reviewer 1 for his (or her) useful comments. We completely agree with your comments. So, we revised our manuscript according to your comments.
Title;
“Rat” was added in the title to show that this study examined the sites of Epo production in the rat body,
Abstracts;
We added that the data were obtained from rats.
Introduction
We list several reports regarding the effects of ROX on CKD patients with renal anemia with or without hemodialysis, and on Epo production in experimental conditions. ROX showed erythropoietic effects in patients with CKD, the same as ESA. In animal studies, Kato et al. reported the increase of Epo mRNA in the liver by TP0463518, a PHD inhibitor (ref. 29). However, moderate hepatic impairment did not affect erythropoietic effects of ROX, suggesting the lack of hepatic Epo production by ROX (ref. 31). The lack of Epo protein determination is the reason for the unclear role of liver for ROX-stimulated Epo production. We have shown that ROX caused Epo production not in liver but in kidney, despite ROX increasing Epo mRNA expression in the liver, the same as reported by Kato et al. We were the first to show the detection of Epo protein by Western blotting without prepurification of the samples. We further reported the increase of the detection limit by enzymatic deglycosylation. Our report using deglycosylated Epo production is the first one. We have confirmed that the bands with or without deglycosylation contain Epo by LC/MS. There have been no reports investigating Epo protein production by ROX.
Epo concentration in blood has been evaluated either ELISA of CLEIA. Many companies provide the measurements of blood Epo concentration. SRL and LSI provide the measurements of blood Epo in Japan.
We cut the description of the results from the Introduction according to your suggestions.
Results
The presentation of the results was largely changed by your comments as follows: Comparison of glycosylated and deglycosylated Epo protein by Western blotting (Fig. 1), Epo production sites among organs by ROX (Fig. 2) and by hypoxia (Fig. 3), Epo mRNA and protein detection in the kidney and liver (Fig. 4), immunohistochemistry (Fig. 5) and plasma Epo concentration (Fig. 6). Old Figures 3 and 4, and old Figure 5 and 6 were combined. The figure legends were shortened.
- Plasma Epo concentration:
Epo concentration in blood can be measured in either serum or plasma; there is no difference. There is also no difference between plasma and serum for detection by Western blotting. Since plasma was more useful to determine hormone concentration in blood such as plasma renin activity or plasma aldosterone concentration, we collected plasma from the rats.
Old Figure 2e was moved to Figure 1 to explain the method of deglycosylation.
The description of the results in the figure legends was deleted.
- Effects of ROX on Epo-related mRNA expression in kidney and liver
Presentation of statistical methods was deleted.
Since we have not reported the details of the severe hypoxia-induced increase of EPo mRNA expression in the liver in our previous report other than the figure, we showed the details of the results in this manuscript. We have used net Epo/b-actin mRNA expression by the liver and the kidney. The maximum value of liver Epo/β-actin mRNA expression was small (1.5%) compared with the maximal value of Epo/β-actin mRNA by the kidney (1.5%). In this report, Epo/β-actin mRNA expression was presented separately in kidney and liver, and hypoxia-induced Epo mRNA expression in the liver became apparent. If you think this presentation is troublesome, we can delete the description.
- Western blot analysis of ROX-induced Epo production in kidney and liver
The band intensity by Western blot was normalized to β-actin.
The results of ROX-induced Epo mRNA and protein expression were combined.
- Western blot analysis of ROX-induced Epo production in the body
The results in male and female were combined. The bar graphs in Figure 7 were combined with the results of Western blot in new Figure 2. After measuring Epo protein expression, the membrane was stripped (stripping solution, Wako, RR39LR, Tokyo, Japan) and reprobed with the antibody against β-actin (MBL, M177-3, Tokyo, Japan) for the normalization of the band. This description was added to the Materials and Methods.
- Epo production during severe hypoxia (in vivo)
The results in male and female were combined. The bar graphs in Figure 7 were combined with the results of Western blot in new Figure 3.
- Immunohistochemistry (IHC) of Epo production by the kidney
The description of the results by immunohistochemistry was moved to before the figure.
The discussion
The description already presented in the Introduction was deleted.
The explanation of the deglycosylated Epo was added to the Discussion. The description regarding our previous reports was shortened. The data in this report but not in previous studies were mainly used in the discussion.
The description regarding deglycosylation was moved to Materials according to your comments.
Materials
The group and numbers of rats used was described in this section.
1: CLEIA and ELISA: We asked SRL about the differences in CLEIA and ELISA. However, the details of the methods cannot be opened by SRL.
2: We used “Real-Time Quantitative RT-PCR” as you have suggested.
3: CT values in Real-Time Quantitative RT-PCR was as follows:
b-actin: 21-26 blank; undetermined
Epo: 24-36 blank, undetermined
HIF2a: 26-33 blank. undetermined
HIF1a: 25-32 blank. undetermined
PHD2: 25-31 blank. undetermined
4: We used the 2 – delta delta CT method described by Livak KJ et al. in this study and our previous studies. We cited the following paper and modified the description of the methods.
Livak KJ, Schmittgen TD. Analysis of relative gene expression data using real-time quantitative PCR and the 2(-Delta Delta C(T)) Method. Methods. 2001 Dec;25(4):402-8.
5: Our samples were divided into two: one is without PNGase and the other is with PNGase. Both samples were incubated in water bath for 15-29 hr at 37°C. We added a description of the samples.
- We used 0.45μm PVDF membrane in our Western blot analysis.
‘7: The molecular weight marker we used was PageRuler Prestained Protein Ladder from Thermo Scientific (26616). The marker shows the bands at 180, 130, 95, 72, 55, 43, 34, 26, 17 and 10 kDa.
8: β-actin was used as reference protein for tissue normalization. After measuring Epo protein expression, the membrane was stripped (stripping solution, Wako, RR39LR, Tokyo, Japan) and reprobed with the antibody against β-actin (MBL, M177-3, Tokyo, Japan) for the normalization of the band.
9: Plasma Epo concentration was not normalized. Just plasma or serum was used for the determination.
The software used for the densitometric analysis was Multi Gauge in LAS 4000 (Fujifilm, Tokyo, Japan)
The transfer of the protein to PVDF membrane was performed after SDS-PAGE. The membrane was sandwiched between thick filter papers and then the transfer of the protein was performed with 120mA constant current for 60-90 min. Denaturation of the samples were done before SDS-PAGE by heating to 95°C for 5 min.
Reviewer 2 Report
1.The significance of the research in introduction is not fully elucidated and should be combined with the clinical significance. Their goal is only developed a novel assay ?
2.Some pictures are of poor quality
Author Response
Answers to the Reviewer 2
We thank Reviewer 2 for his (or her) useful comments. We completely agree with your comments.
1: We added the description regarding the clinical and basic studies by the treatment by ROX in the Introduction. Our experiments were an animal study. We did not examine the long-term effect of ROX. However, the clinical effectiveness of ROX has been reported and established in many clinical studies, so we cited such studies. Our Western blot can determine Epo protein in tissues and enzymatic deglycosylation increased the assay sensitivity.
2: We modified the figures for better understanding of the results. The order of the figures was changed according to the comments by Reviewer 1.
β-actin was used as reference protein for tissue normalization. After measuring Epo protein expression, the membrane was stripped (stripping solution, Wako, RR39LR, Tokyo, Japan) and reprobed with the antibody against β-actin (MBL, M177-3, Tokyo, Japan) for the normalization of the band.
Round 2
Reviewer 1 Report
Dear authors,
You took into account all the comments, correctly placed the emphasis and significantly improved the manuscript. I have only minor comments below.
A note about the duplication of the description of the results in the legends to the figures remains. There is no need to give a detailed description in the legend. You have already mentioned this in the text. All that needs to be indicated under the figure is the decoding of the samples and their belonging. For example, the legend for Fig. 1. ROX-treated kidney lysates were deglycosylated with and without PNGase and were examined by two-103 fold serial dilution. Deglycosylated samples showed a lower detection limit than glycosylated samples. Same 104 numbers (lanes 1-5) show same amount of glycosylated and deglycosylated Epo. You only need to specify decoding of the abbreviations, such as rRat Epo and degEpo. Otherwise, the legend again repeats the results described above in the text. The same applies to figures 2,3,4, 6.
All figures should be strictly placed immediately after their description in the text.
To Plasma Epo concentration
Please correct the numbers of figures in the text (highlighted in yellow)
Plasma Epo concentrations in control and ROX-treated rats are shown in Fig. 1a. ROX (50 mg/kg) 249 significantly increased plasma Epo concentration from 1.2 ± 0.1 to 1,072 ± 333 mIU/ml (n=5-6, p<0.001). 250 Plasma Epo expression was also estimated by Western blot. ROX increased plasma Epo concentration (1.4, 251 2690 and 3320 mIU/ml in control, R5 and R10, respectively). By Western blot of plasma Epo, ROX dose-252 dependently increased both the glycosylated and deglycosylated Epo protein (Fig. 1b).
To Discussion. At the end of the Discussion. Please, indicate the abbreviated REP (renal erythropoietin producing cells) after it is deciphered, since this term is used later in the text without decoding.
To Materials. Add to the description - 0.45 micron PVDF membrane thickness; the molecular weight marker used was PageRuler from Thermo Scientific (26616) and software - LAS 4000 (Fujifilm, Tokyo, Japan).
Author Response
Answers to the Reviewer 1
We thank Reviewer 1 for his (or her) useful comments. We revised our manuscript according to your comments.
Results
Figure legends
We took the description of the results in the figure legends in figures 1, 2,3,4 and 6 according to your comments.
Figures are placed immediately after the description in the text.
Line 153-154; The sentence was slight modified for the better understanding.
Line 154: PHD mRNA was changed to PHD2 mRNA
Plasma Epo concentration
We corrected the figure number in the results of plasma Epo concentration from Fig.1 to Fig.6.
Discussion
Full spelling of REP cells (renal erythropoietin producing cells) was added to the discussion.
Line 223-224, 227-228; The sentence was modified for the better understanding.
Materials
The thickness of PVDF membrane 0.45μm was added in the text.
The name of the molecular weight marker used (PageRuler) was added in the text.